# Metabolite Profiling of Wheat Response to Cultivar Improvement and Nitrogen Fertilizer

**DOI:** 10.3390/metabo13010107

**Published:** 2023-01-09

**Authors:** Fulin Zhao, Yifan Wang, Jiayu Hu, Shaolei Shi, Hongyan Zhang, Yang Wang, Youliang Ye

**Affiliations:** 1Agricultural Green Development Engineering Technology Research Center, College of Resources and Environment, Henan Agricultural University, Zhengzhou 450002, China; 2Dengzhou Agricultural Technology Extension Center, Dengzhou 474150, China; 3Majorbio Company, Ai Di Sheng Rd., Zhangjiang Hi-Tech Park, Shanghai 201203, China

**Keywords:** yield, grain quality, metabolome, wheat release year, nitrogen nutrient

## Abstract

Both genetic improvement and the application of N fertilizer increase the quality and yields of wheat. However, the molecular kinetics that underlies the differences between them are not well understood. In this study, we performed a non-targeted metabolomic analysis on wheat cultivars from different release years to comprehensively investigate the metabolic differences between cultivar and N treatments. The results revealed that the plant height and tiller number steadily decreased with increased ears numbers, whereas the grain number and weight increased with genetic improvement. Following the addition of N fertilizer, the panicle numbers and grain weights increased in an old cultivar, whereas the panicle number and grain number per panicle increased in a modern cultivar. For the 1950s to 2010s cultivar, the yield increases due to genetic improvements ranged from −1.9% to 96.7%, whereas that of N application ranged from 19.1% to 81.6%. Based on the untargeted metabolomics approach, the findings demonstrated that genetic improvements induced 1.4 to 7.4 times more metabolic alterations than N fertilizer supply. After the addition of N, 69.6%, 29.4%, and 33.3% of the differential metabolites were upregulated in the 1950s, 1980s, and 2010s cultivars, respectively. The results of metabolic pathway analysis of the identified differential metabolites via genetic improvement indicated enrichment in 1-2 KEGG pathways, whereas the application of N fertilizer enriched 2–4 pathways. Our results provide new insights into the molecular mechanisms of wheat quality and grain yield developments.

## 1. Introduction

Wheat (*Triticum aestivum* L.) is one of the world’s most widely cultivated crops and plays a critical role in ensuring national food security [1]. The arable land in China accounts for ~7% of the world’s total; of the crops produced on this land, wheat production accounts for 20.30% and feeds 22% of the global total [2]. In limited cultivated farmlands, food security must be ensured through continuous increases in grain yields. Since 1949, China’s wheat cultivars have undergone 4–6 large-scale replacements [3]. Prior to the 1980s, yield increases, stress resistance, and regional adaptability were the primary drivers of the replacements of wheat cultivars [4]. However, with economic development, improvements in individual living standards, and changes in dietary structure, wheat production structures were adjusted to adapt to the market, particularly flour enterprises that placed higher requirements on wheat quality [5]. Several studies have reported the evolution of wheat quality and yields during cultivar improvement [6,7]. Due to differences in the cultivar and ecological environments, the conclusions of these different studies are not entirely consistent.

Historical sets of various cereal crops were extensively evaluated in earlier studies to improve the data regarding genetic advances in grain yield and the associated physiological traits [8]. Two major breakthroughs in breeding contributed to the dramatic increase in average wheat yields over the last half-century. The first was the Green Revolution, which saw the rise of dwarf plant breeding, whereas the second was the distant hybridization between wheat and its wild relatives [9,10]. In addition to genetic improvements, N fertilizer plays a decisive role in agricultural production [11]. The proper application of N fertilizer increases the absorption areas and activities of crop root systems to promote the emergence of productive wheat tillers [12,13]. According to FAO data, ~55% of the increase in grain production in developing countries has been derived from chemical fertilizers. In the early 1970s, China began to apply large quantities of N fertilizers. With the rapid development of the fertilizer industry, China has become the largest consumer and producer of N fertilizer in the world [14]. Nitrogen directly or indirectly affects many aspects of the metabolism, growth, and development of plants, and is the main element that impacts the quality and yields of wheat [15]. Researchers have previously described the kinetics of N for improving crop quality and yield from the perspectives of agronomic traits, photosynthetic physiology, ecological effects, and nutrient absorption and transport [16,17]. 

Metabolomics has become a powerful tool in the post-genomics era, which enables the exploration of different facets of biological and physiological changes caused by environmental or genetic perturbations [18]. Xv et al. [19] compared the predictive effects of various omics on crop yields, with the results showing that the metabolome produced the highest predictability followed by the transcriptome and genome. Metabolites have been used to predict plant phenotypes and quality [20,21]. For example, crop engineering produced increased levels of phenylpropanoids antioxidants to be used to promote the health of fruit and vegetable crops. Metabolomics is an important addition to the tools that are currently employed in genomics-assisted selection for crop improvement. Furthermore, because metabolites are last in the omics cascade (and are relatively close to the phenotype), they are a reliable tool for investigating abiotic stress responses in plants [22]. In terms of N-deficiency tolerance, altered metabolites typically indicated that most of the amino acid content decreased, whereas most phenylpropanoids and organic acids increased in plant leaves [23]. Furthermore, glutamate and aspartate levels were recognized as the best indicators of the stresses related to N limitations [24].

Although genetic improvements and fertilization practices are designed to enhance crop sources and sink capacities, whether the metabolites and metabolic pathways they affect are similar is not yet known. As such, in this study, we applied non-targeted metabolomic analysis to compare cultivars between different release years under N-deficient and adequate-N treatments. Our elucidation of the yield-increasing mechanisms of these two strategies based on metabolic profiles will assist with the generation of high-quality and high-yield crops through metabolic engineering and the precise application of exogenous substances in the future.

## 2. Materials and Methods

### 2.1. Experimental Site 

We conducted this study in Xinxiang, Henan(35°18′ N, 113°95′ E), in central China. During the wheat-growing seasons, the total precipitation and the mean temperature were 156.4 mm and 10.8 °C (2019–2020) and 149.5 mm and 9.7 °C, respectively (2020–2021). Prior to the investigation, we extracted soil samples from the upper 30 cm layer for chemical analyses. The soil type of the test site was fluvo-aquic and sandy, and the pH of the cultivated soil layer was 6.8, with an organic matter content of 15.3 g·kg^−1^, which contained 1.1 g·kg^−1^ total N, 19.8 mg·kg^−1^ available P, and 88.0 mg·kg^−1^ available K.

### 2.2. Plant Materials and Experimental Design

We used seven historical wheat cultivars released from the 1950s to the 2010s in this study, all of which were cultivated as winter wheat in a large-scale area in the North China Plain over the 60-year periods (Table 1). A total of 165 kg seeds ha^−1^ were mechanically sown on 3 October 2019 and 6 October 2020 with the dimensions of each plot being 6.0 × 8.0 m. We arranged the experiment with a randomized complete block design with three replications, and we used two N treatments: 0 and 180 kg N ha^−1^. We applied N (urea) at two intervals, with 50% at the basal and 50% at the jointing stage (~160 days after sowing (DAS). We applied phosphorus (90 kg P_2_O_5_ ha^−1^) (as calcium superphosphate) and potassium (90 kg K_2_O ha^−1^) (as potassium chloride) as the basal dose. We applied the basal fertilizer to the ground following manual broadcasting, and we applied N topdressing via side-dressing. We employed the wheat–maize rotation system in the field sites during the experimental period. The fertilization rate for the maize-growing season was the same as that for the wheat-growing season in each plot. Wheat typically requires three rounds of irrigation (each at 75 mm water), whereas maize requires only two rounds of irrigation (each at 100 mm water). We intensively controlled weeds, diseases, and insects during the entire growing season to avoid yield losses.

### 2.3. Plant Measurements and Analysis

#### 2.3.1. Grain Quality and Yield Measurements

At harvest, we measured the wheat grain yield for 6 m^2^ areas in each plot using hand harvesting and machine threshing. We adjusted the grains to a moisture content of 0.13 g H_2_O g^−1^ fresh weight. Once we air-dried the grains, we ground them into flour using a laboratory mill instrument (CD1, Chopin Technologies, Villeneuve-La-Garenne, France). Next, we measured the crude protein and wet gluten using the AACC No.39-11 and AACC No.38-12 standards, respectively. We measured the sedimentation value following the AACC No.58-61B standards, via the fungal falling number method (FN1000, Perten Instruments, Hagersten, Sweden).

#### 2.3.2. Metabolite Extraction

We accurately weighed 50 mg of fresh leaves and extracted the metabolites using a 400 µL methanol: water (4:1, *v*/*v*) solution. We allowed the mixture to settle at −20 °C and treated it with a high-throughput tissue crusher Wonbio-96c (Shanghai Wanbo Biotechnology Co., Ltd., Shanghai, China) at 50 Hz for 6 min, which was followed by vortexing for 30 s and ultrasound at 40 kHz for 30 min at 5 °C. We stored the samples at −20 °C for 30 min to precipitate the proteins. Following centrifugation at 13,000× *g* at 4 °C for 15 min, we carefully transferred the supernatant to sample vials for LC-MS/MS analysis.

#### 2.3.3. LC-MS Conditions for Non-Targeted Metabolomic Analysis

As part of the system conditioning and quality control process, we prepared a pooled quality control sample (QC) by mixing equal volumes of all samples. We prepared and tested the QC samples in the same manner as the analytic samples. This assisted with representing the entire sample set, which we injected at regular intervals (every eight samples) to monitor the stability of the analysis.

We chromatographically separated the metabolites using a Thermo UHPLC system equipped with an ACQUITY UPLC HSS T3 column (100 mm × 2.1 mm i.d., 1.8 µm) (Waters, Milford, CT, USA). The mobile phases consisted of 0.1% formic acid in water (solvent A) and 0.1% formic acid in acetonitrile: isopropanol (1:1, *v*/*v*) (solvent B). We changed the solvent gradient according to the following conditions: From 0 to 3 min, 95% (A): 5% (B) to 80% (A): 20% (B); from 3 to 9 min, 80% (A): 20% (B) to 5% (A): 95% (B); from 9 to 13 min, 5% (A): 95% (B) to 5% (A): 95% (B); from 13 to 13.1 min, 5% (A): 95% (B) to 95% (A): 5% (B), from 13.1 to 16 min, 95% (A): 5% (B) to 95% (A): 5% (B) for equilibrating the systems. The sample injection volume was 2 μL and we set the flow rate to 0.4 mL/min. We maintained the column temperature at 40 °C. During the analysis period, we stored all samples at 4 °C.

We collected mass spectrometric data using a Thermo UHPLC-Q Exactive Mass Spectrometer equipped with an electrospray ionization (ESI) source operating in either positive or negative ion mode. We established the optimal conditions as follows: Aux gas heater temperature, 425 °C; sheath gas flow rate 50 psi; aux gas flow rate 13 psi; ion-spray voltage floating (ISVF), −2800 V in negative mode and 3500 V in positive mode; normalized collision energy, 20-40-60 V rolling for MS/MS. We acquired data acquisition in data-dependent acquisition (DDA) mode, and we performed the detection over a mass range of 70–1050 m/z.

#### 2.3.4. Data Processing for Nontargeted Metabolomic Analysis

Following UPLC-TOF/MS analyses, we transferred the raw data to Progenesis QI 2.3 (Nonlinear Dynamics, Waters, Milford, CT, USA), where we performed the identification of peaks and their alignment. The applied preprocessing resulted in a data matrix that encompassed the peak intensity, mass-to-charge ratio (*m*/*z*) values, and retention time (RT). We retained the metabolic elements if at least 80% were identified in any of the sample sets. Following filtering, we assigned minimal values to the metabolites of certain samples when they were below measurable levels, and we sum-normalized each metabolic element. We employed an internal standard for the QC of data to facilitate reproducibility, where we removed metabolic features with a QC relative standard deviation (RSD) of >30%. Subsequent to normalization and designation, we performed statistical analysis on log-transformed data to assess any major variations in metabolite values among equivalent groups. We determined the mass spectra of these metabolic elements by using the precise mass, MS/MS fragment spectra, and differences in isotope ratios by surveying dependable biochemical databases (e.g., human metabolome database (HMDB) (http://www.hmdb.ca/, accessed on 24 November 2022) and the Metlin database (https://metlin.scripps.edu/, accessed on 24 November 2022). Definitively, the mass tolerance between the quantified *m*/*z* values and the exact mass of the elements of interest was ±10 ppm. For MS/MS confirmation metabolites, we deemed only those with MS/MS fragment scores of >30 to be positively identified. Otherwise, we only tentatively assigned the metabolites.

### 2.4. Statistical Analysis

We selected statistically significant metabolites among the sample groups with VIP values of >1 and *p* values of <0.05. We selected 28,534 differential peaks, including 12,599 peaks in ESI+ and 15,935 peaks in ESI-. We summarized and mapped the differential metabolites between two groups into their biochemical pathways through metabolic enrichment and pathway analyses based on a database search (KEGG, http://www.genome.jp/kegg/, accessed on 24 November 2022). We classified these metabolites according to the pathways in which they were engaged or the functions they performed. Enrichment analysis typically indicates whether or not a group of metabolites exists in a function node. The principle is that the annotation analysis of a single metabolite develops into an annotation analysis of a group of metabolites. We used scipy. stats (Python packages) (https://docs.scipy.org/doc/scipy/, accessed on 24 November 2022) to identify statistically significantly enriched pathways using Fisher’s exact test.

## 3. Results

### 3.1. Genetic Improvements and N Application for Increased Cultivar Quality and Yield

The results of two-way ANOVAs showed that the grain yield, crude protein, wet gluten, and sedimentation value were significantly affected by cultivar (C) and nitrogen (N) in these two growing seasons (Figure 1A–H). Additionally, the effects of C and N interactions on grain yield and quality were highly significant. For both growing seasons, the cultivar yields increased with progressing release years. Under the -N treatments, the grain yields of newer cultivars were significantly higher than those of older cultivars (Figure 1A,B). The yields from the 1950s–1980s cultivars increased over time during the two growing seasons, while those from the 1980s–2010s did not change (2019–2020) or even decreased (2020–2021). Following N fertilization, the grain yields of all cultivars significantly increased at rates that ranged from 33.8% to 65.1% (2019–2020) and 19.1% to 81.6% (2020–2021). Distinct from the modified yield, grain-quality-related indicators did not exhibit a uniform trend (Figure 1C–H). The crude protein and sedimentation values of the 1980s cultivar were significantly higher than those of other cultivars. Notably, the application of N fertilizer still markedly improved the grain quality, crude protein, sedimentation value, and wet gluten by 19.6%, 20.5%, and 44.7%, respectively, on average during the two growing seasons.

From the 1950s to the 2010s, the agronomic plant heights and panicle numbers steadily decreased and the grain numbers and weights increased (Table 2). The results of two-way ANOVA indicated that all agronomic traits were affected by C or N, and we found significant C × N interactions in the number of panicles and grains per panicle. Following N fertilization, plant heights and panicle numbers significantly increased, and the responses of grain numbers and weights varied for the cultivars of different eras. The grain weights of the 1950s, 1960s, and 1980s were considerably increased by the addition of N, whereas we noted few changes in the other cultivars.

### 3.2. Effects of Genetic Improvement and N Application on Metabolic Profiles

The PLS-DA plot suggested that the metabolic profiles differed between the various wheat cultivars, with the results indicating that leaf metabolites were clustered by the release era (Figure 2). The positive and negative metabolites were significantly different between the cultivars under the in −N and +N treatments. Furthermore, the loading plot from the analysis showed a clear separation in metabolites between the -N and +N groups for both positive and negative modes (Figure 3).

### 3.3. Overview of Altered Metabolites under Genetic Improvement and N Application Groups

Compared with the oldest cultivar (the 1950s), 78 metabolites in the 1980s cultivar were upregulated and 93 were significantly downregulated (Table 3). The main types of regulated metabolites were lipids and lipid-like molecules (42.50%), which indicated the ratio of these substances to the total differential metabolites. Additional constituent metabolites included phenylpropanoids and polyketides (20.62%), organic oxygen compounds (16.12%), and organoheterocyclic compounds (9.38%). Compared with the 1980s cultivar, 33 metabolites in the 2010s cultivar were upregulated and 42 were significantly downregulated. The main types of regulated metabolites were lipids and lipid-like molecules (30.14%), followed by phenylpropanoids and polyketides (27.40%), organic oxygen compounds (19.18%), and organic acids and derivatives (8.22%). We found 36 shared differential metabolites between the 2010s vs. 1980s and 1980s vs. 1950s groups, where most of them were attributed to lipids and lipid-like molecules, phenylpropanoids, and polyketides (Figure 4A; Appendix A).

Distinct from genetic improvements, the application of N fertilizer regulated fewer metabolites. We identified 23 altered metabolites (16 upregulated and 7 downregulated) for the 1950s cultivar, 68 (20 upregulated and 48 downregulated) for the 1980s cultivar, and 54 (18 upregulated and 36 downregulated) for the 2010s cultivar (Table 2). The main types of regulated metabolites for the +N vs. −N groups for the 1950s cultivar included phenylpropanoids and polyketides (30.00%), organoheterocyclic compounds (25.00%), lipids and lipid-like molecules (20.00%), and organic acids and derivatives (10.00%). The primary types of regulated metabolites for the +N vs. −N groups for the 1980s cultivar were organic acids and derivatives (33.9%), lipids and lipid-like molecules (32.20%), organic acids and derivatives (15.25%), and organoheterocyclic compounds (11.86%). The main types of regulated metabolites for the +N vs. vs. −N group in the 2010s cultivar were lipids and lipid-like molecules (44.90%), phenylpropanoids and polyketides (20.41%), organic oxygen compounds (14.29%), and organic acids and derivatives (10.20%). We identified only one shared differential metabolite (5′-carboxy-gamma-chromanol) between the above three N treatment groups (Figure 4B; Appendix A).

### 3.4. Altered Metabolic Pathways in Genetic Improvement and N Application Groups

The results of our metabolic pathway analyses of the identified differential metabolites indicated differences in one carbon pool via folate and tryptophan metabolism between the 1980s and 1950s cultivars, and only one pathway difference in aflatoxin biosynthesis between the 2010s and 1980s cultivars (Figure 5A,B).

We identified two KEGG pathways (anthocyanin and flavonoid biosynthesis) enriched by differential metabolites following the addition of N fertilizer in the 1950s cultivar; two enriched pathways (arginine and proline metabolism and one carbon pool by folate) in the 1980s cultivar; and four enriched pathways (pyrimidine metabolism, phenylpropanoid biosynthesis, flavonoid biosynthesis, and stilbenoid, diarylheptanoid, and gingerol biosynthesis) in the 2010s cultivar (Figure 6A–C).

### 3.5. Relationships between Metabolic Profiles and Quality and Yield in Genetic Improvement and N Application Groups

A heatmap showed that seven lipids and lipid-like molecules, three organic acids and derivatives, two phenylpropanoids and polyketides, one organic oxygen compound, and one organoheterocyclic compound of the VIP top 30 differential metabolites positively correlated with yields in the 1980s and 1950s cultivar groups. Furthermore, four lipids and lipid-like molecules, three organic oxygen compounds, two benzenoids, two phenylpropanoids and polyketides, one organic acid and derivative, and one other negatively correlated with yield. Except for phenyllactic acid, graecnin E, trihydroxyoxane, and trehalose 6-phoshate, most of the differential metabolites related to yields were related to quality indicators (Figure 7A). When the cultivar was updated from the 1980s to the 2010s, we found no substances that affected the wet gluten content. However, three phenylpropanoids and polyketides, one lipid and lipid-like molecule, one organic acid and derivative, one organic oxygen compound, and one organoheterocyclic compound of the VIP top 30 differential metabolites positively correlated with yields, crude proteins, and sedimentation values. Furthermore, two lipids and lipid-like molecules, two organic oxygen compounds, and one benzenoid negatively correlated with yield, crude protein, and sedimentation values (Figure 7B). Additionally, a small number of differential metabolites was related to either grain yield or quality.

The heatmap revealed many more differential metabolites of the VIP top 30 in the +N vs. −N groups of the 1950s cultivar, which positively correlated with grain yields or quality, whereas more differential metabolites in the +N vs. −N groups of the 1980s and 2010s cultivars negatively correlated with grain yield or quality (Figure 8A–C). In the 1950s cultivar, two organoheterocyclic compounds; one nucleoside, nucleotide, and analogue; and one other of the VIP top 30 differential metabolites positively correlated with grain yield and quality following the application of N, whereas two phenylpropanoids and polyketides, one organoheterocyclic compound, and one lipid and lipid-like molecule negatively correlated with grain yield and quality (Figure 8A). In the 1980s cultivar, only one organic oxygen compound positively correlated with grain yield and quality following the addition of N, whereas one organic oxygen compound, one phenylpropanoid and polyketide, and one other were negatively correlated with grain yield and quality. Furthermore, the majority of differential metabolites negatively correlated with yield but they were unrelated to grain quality (Figure 8B). In the 2010s cultivar, three lipid-like molecules, two organic acids and derivatives, and one other positively correlated with grain yield and quality after the addition of N, whereas five lipid-like molecules, two organic oxygen compounds, and two others negatively correlated with grain yield and quality (Figure 8C).

## 4. Discussion

### 4.1. Effects of Cultivar Improvements on Wheat Yield and Quality Based on Metabonomics

In this study, our results revealed that the modern cultivars performed better than their older counterparts under both N fertilization levels. Over a period of 60 years, genetic improvements increased crop yields by up to 98.2% (2019–2020) and 96.7% (2020–2021), whereas the application of N fertilizer increased yields by up to 65.1% (2019–2020) and 81.6% (2020–2021) (Figure 1A,B). The yield increases achieved through genetic improvements were primarily attributed to the continuously increased spikelet numbers and grain weights, which were accompanied by a decrease in panicle numbers (Table 1). These results are akin to those of Giunta et al. [7], with the only difference being that the grain weight of a modern wheat cultivar in Italy was 10% less than that of the old cultivar. Furthermore, our results align well with those from a study on rice (both rice and wheat belong to Poaceae) by Zhu et al. [25].

We investigated the metabolic profiles of different wheat cultivars based on an untargeted metabolomics approach. The notable distinction between the three clusters under the -N or +N treatments suggested a significant difference in the metabolic composition between groups (Figure 2). Additionally, the 1980s vs. 1950s groups produced more than twice as many differential metabolites as the 2010s vs. 1980s groups (Table 3). This meant that changes in the attributes of the cultivars from the 1950s to 2010s were more prominent during the initial 30 years in northern China.

Metabolites are closely related to crop yield and quality, and metabolomic strategies can be applied to improve crop function, enhance nutritional quality, and increase grain yield [26,27]. We found significant disparities in the metabolic elements between the 1980s and 1950s cultivars, primarily focused on lipids and lipid-like molecules, phenylpropanoids and polyketides, organic oxygen compounds, organic oxygen compounds, and organic acids and derivatives (Table 3 and Appendix A). Most organic acid levels were higher in the 1980s cultivar, whereas most lipids (prenol lipids and fatty acyl) were higher in the 1950s cultivar. Organic acids are generally intermediate respiration and photosynthesis products in higher plants, which are essential for ammonia assimilation and amino acid synthesis [23]. 

The results of the KEGG pathway enrichment analysis confirmed that the down-regulation of N-acetylserotonin in amino acid metabolism delayed tryptophan degradation (Figure 5). Thus, more peptides accumulated in the 1980s cultivar, among which arginyl-proline, leucyl-phenylalanine, and threoninyl-proline were significantly positively related to grain yield and quality (Figure 7). Typically, these small peptides play critical signaling roles in plants [28]. Of the many decreased lipids levels in the 1980s cultivar, four lipids (methoxyestrone, bornanone glucoside, capsidio, and curcumadiol) were significantly negatively related to grain yields and quality. No relevant report suggests that methoxyestrone and bornanone glucoside impacted plant growth; however, capsidio and curcumadiol may improve the protective enzyme activities and fungal disease resistance of plants [29,30].

Although we found fewer differences in the metabolic elements between the 2010s and 1980s cultivars, we noted more similarities in the tillers, plant height, yield, quality, and other traits. The levels of the majority of organic acids were higher in the 1980s cultivar, whereas most of the lipid levels (prenol lipids and fatty acyls) were higher in the 2010s cultivar. Among these organic acids, dynorphin B was significantly positively related to grain yields and quality. For this substance, researchers previously found that it had an analgesic effect in animals; however, it had not been reported in plants [31]. The KEGG pathway results showed that the differential metabolites were enriched in aflatoxin biosynthesis, and we found some aflatoxin G in the 2010s cultivar (Figure 5). As a potent carcinogen, it might also inhibit the growth of plants, though no inhibitory mechanism has been reported to date [32].

### 4.2. Effects of N Fertilizer Application on Wheat Yield and Quality Based on Metabonomics

Higher wheat yield was premised on the accumulation of more aboveground dry matter, whereas adequate N absorption was deemed to be critical for its accumulation [33,34]. In this study, further increasing the yield of modern wheat cultivars through genetic improvements in the absence of N fertilizers appears to be difficult. However, modern cultivars are increasingly dependent on N for high grain yields (Figure 1). Following the application of N, the increased panicle numbers and grain weight production increased in the 1950s and the 1980s, whereas the yield increases of the 2010s cultivar were caused by the increased panicle number and grain number per panicle (Table 1). The increased rate of productive tillers induced by the addition of N was higher in the old cultivars than in the modern cultivar. However, most small tillers in the old cultivar had fewer grains; thus, we observed a reduced grain number per panicle under the +N treatment.

We found a more distinct separation between the -N and +N treatments in wheat leaves via PLS-DA, the results of which suggested an apparent metabolic alteration induced by N deficiencies. Similar findings were also reported in the N-deficient leaves of tomato [35], tea [23], and rice [36]. In the present study, the number of differential metabolites induced by the application of N was far smaller than that caused by genetic improvements (Table 3). Because metabolites are useful predictors of quantitative traits [19], this finding indicated that genetic improvements more strongly impacted agronomic traits than the application of N. Following N fertilization, the numbers of differential metabolites of the 1980s cultivar were the highest, followed by those of the 2010s cultivar, with the 1950s cultivar producing the fewest. Carboxy-chromanol was the only substance that was upregulated after N application for all wheat cultivars, which can suppress colon tumors and colitis in animals; however, no report has been published on its effects on plants [37,38].

For the 1950s cultivar, most differential metabolites were upregulated following the application of N, with only a few differential metabolites being down-regulated (Appendix A). Among them, delphinidin was enriched in two pathways, anthocyanin and flavonoid biosynthesis (Figure 6). N-deficiencies inhibited protein synthesis, which promoted the increased accumulation of carbohydrates for anthocyanin synthesis [39]. For the 1980s and 2010s cultivars, the majority of differential metabolites were downregulated after the application of N, particularly for carbohydrates and carbohydrate conjugates, lipids and lipid-like molecules, and phenylpropanoids and polyketides. The accumulation of carbohydrates and lipids in N-deficient plants increased osmotic pressure, which resulted in the over-accumulation of reactive oxygen species [23]. In the present study, the KEGG pathway enrichment analysis confirmed that the downregulation of chlorogenic acid in phenylpropanoid biosynthesis delayed the degradation of phenylalanine (Figure 8). The phenylpropanoid biosynthesis pathway is involved in plant disease resistance and immunity [40,41].

## 5. Conclusions

Both genetic improvements and the application of N enhanced the yield and quality of wheat. From the 1950s to 2010s cultivars, the yield increase achieved through genetic improvement ranged from −1.9% to 96.7%, whereas that achieved through the application of N ranged from 19.1% to 81.6%. Based on the results of an untargeted metabolomics approach, we found that genetic improvements more strongly induced metabolic modification than N fertilization. Compared with the 1950s cultivar, the 1980s cultivar showed higher relative abundances of most organic acids, and lower relative abundances of most lipids in wheat leaves. Of the subsequent cultivar improvements, the 2010s cultivar showed upregulated relative abundances of most lipids and downregulated relative abundances of most organic acids and organoheterocyclic compounds compared to the 1980s cultivar. Differentially, essentially all of the differential metabolites were upregulated in the 1950s cultivar, whereas many differential metabolites were downregulated, particularly for carbohydrates and carbohydrate conjugates, lipids and lipid-like molecules, and phenylpropanoids and polyketides. These findings strengthen our understanding of the molecular mechanisms that underlie the differences between genetic improvements and N applications to increase wheat grain yield and quality.

## Figures and Tables

**Figure 1 metabolites-13-00107-f001:**
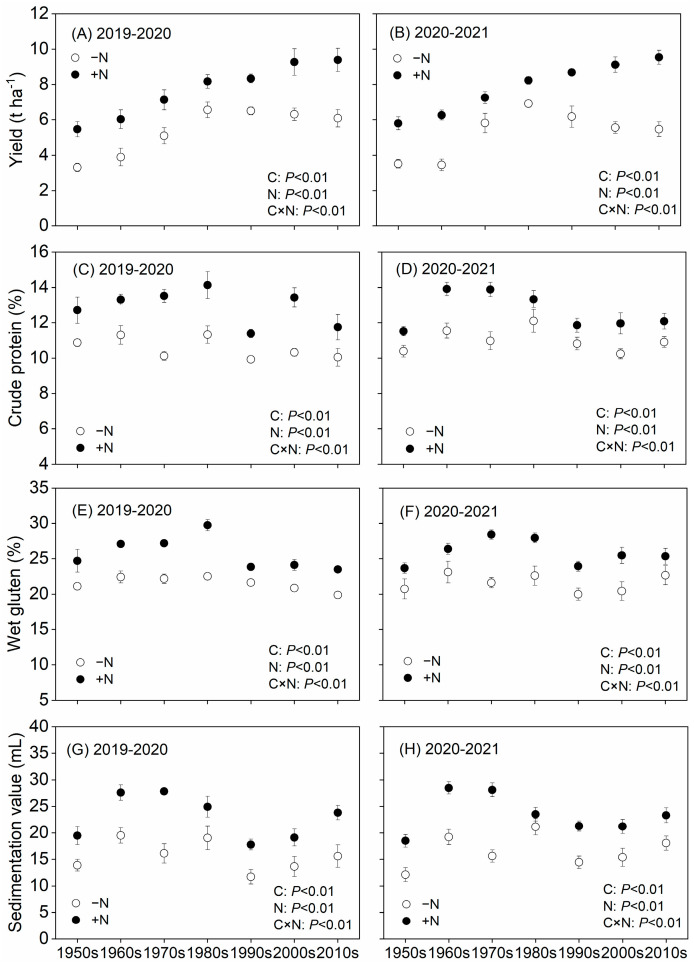
Yields and quality of wheat cultivars from different release years under −N and +N treatments from 2019–2020 (**A**,**C**,**E**,**G**) and 2020–2021 (**B**,**D**,**F**,**H**).

**Figure 2 metabolites-13-00107-f002:**
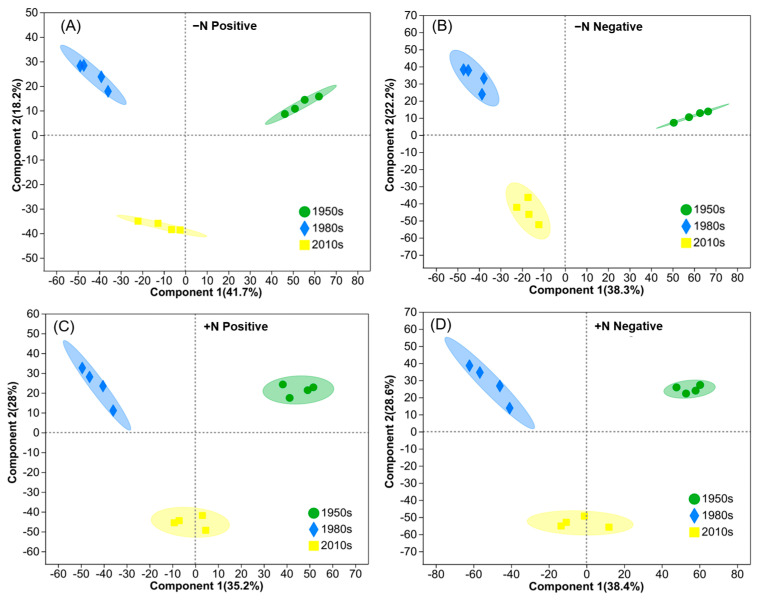
Partial least-squares–discriminant analysis (PLS-DA) of leaf metabolic profiles of different years of released cultivar under −N (**A**,**B**) or +N (**C**,**D**) treatments. Positive and negative modes are shown.

**Figure 3 metabolites-13-00107-f003:**
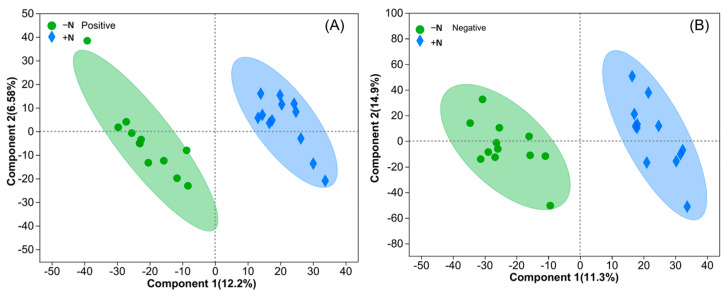
Partial least-squares–discriminant analysis (PLS-DA) of leaf metabolic profiles between the +N and −N treatments: positive (**A**) and negative (**B**) modes.

**Figure 4 metabolites-13-00107-f004:**
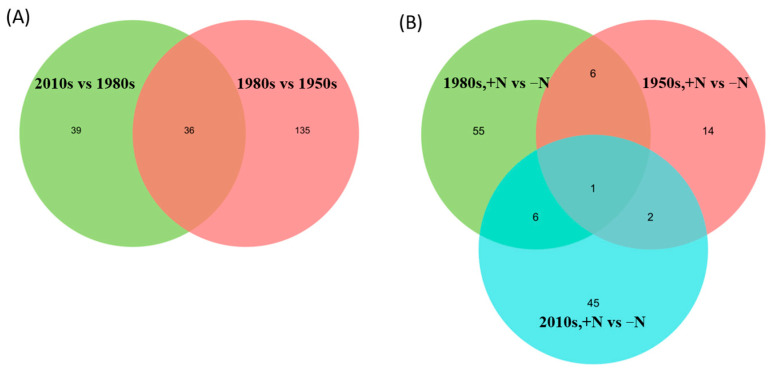
Venn diagram of differential metabolites in cultivar group (**A**) and N treatment group (**B**).

**Figure 5 metabolites-13-00107-f005:**
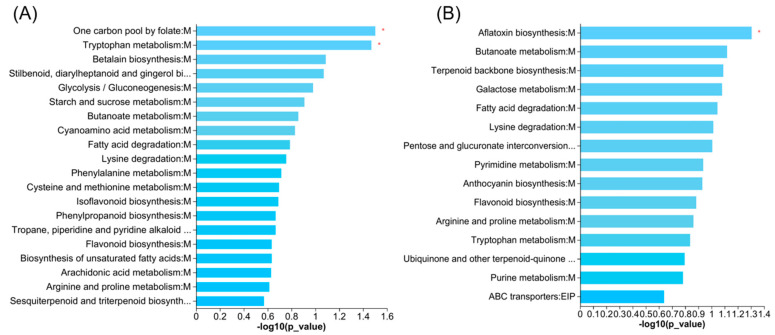
KEGG enrichment maps of different metabolites in 1980s vs. 1950s groups (**A**) and 2010s vs. 1980s groups (**B**). * indicate enrichment with significance at 0.05 level.

**Figure 6 metabolites-13-00107-f006:**
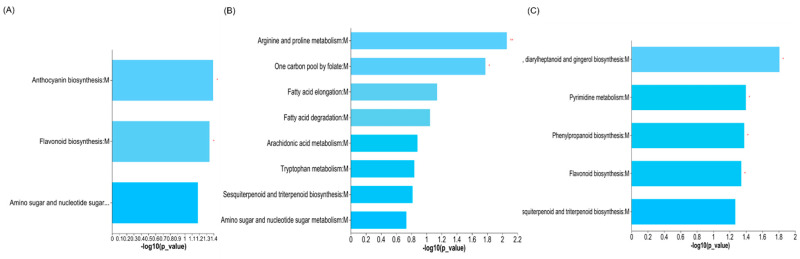
KEGG enrichment maps of different metabolites between +N and −N treatments of 1950s cultivar (**A**), 1980s cultivar (**B**), and 2010s cultivar (**C**). *and ** indicate enrichment with significance at the 0.05 and 0.01 levels, respectively.

**Figure 7 metabolites-13-00107-f007:**
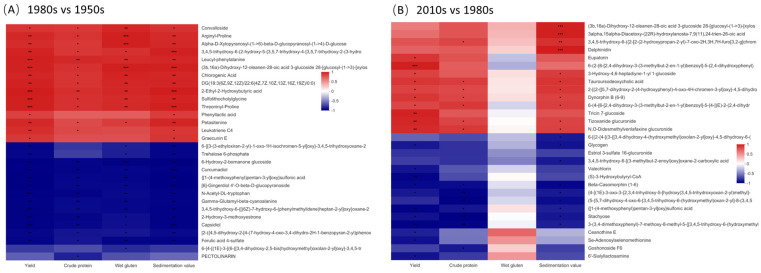
Correlation between metabolics and grain yield and quality in 1980s vs. 1950s groups (**A**) and 2010s vs. 1980s groups (**B**). *, ** and *** indicate the correlation is significant at 0.05, 0.01 and 0.001 levels, respectively.

**Figure 8 metabolites-13-00107-f008:**
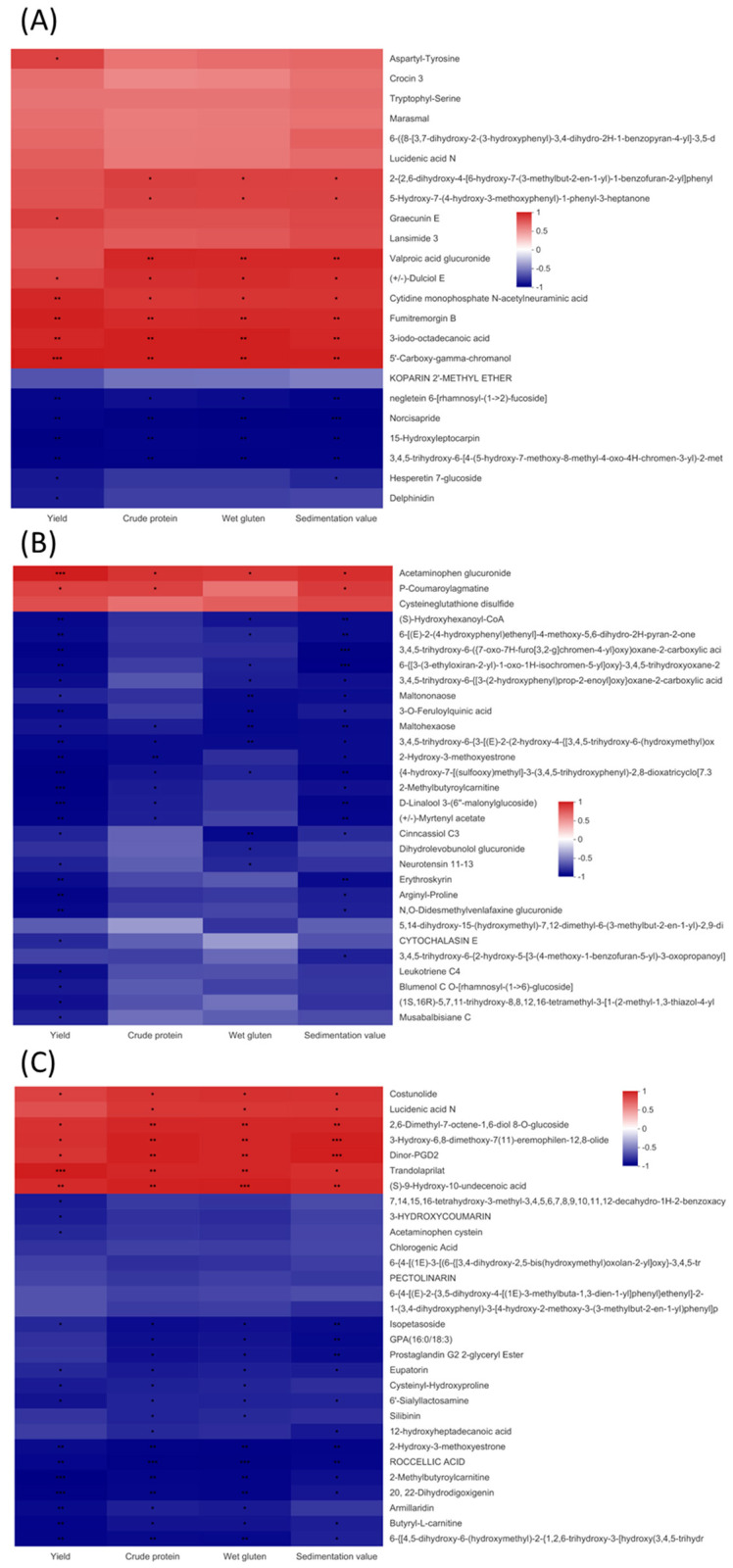
Correlation between metabolics and grain yield and quality in the +N vs. −N groups of 1950s (**A**), 1980s (**B**), and 2010s (**C**) cultivars. *, ** and *** indicate the correlation is significant at 0.05, 0.01 and 0.001 levels, respectively.

**Table 1 metabolites-13-00107-t001:** Designation of seven selected wheat cultivars.

Cultivar	Pedigree/Origin	Release Period
ND2419	Rieti × Wilhelmina//Akagomughi	1950s
BJ8	BM4 × Early Premium	1960s
ZY1	St1472/506	1970s
XY4	ZY4 × ZZ17 × 6609	1980s
BN3217	Funo × NX5//XN39 × XN64 × YD34	1990s
YM2	ZN16 × YM14	2000s
BN207	Z16 × BN64	2010s

**Table 2 metabolites-13-00107-t002:** Agronomic traits of wheat cultivars from different years of release under −N and +N treatments. Significant differences between two N treatments for same cultivar are indicated by ns, no significance; * *p* < 0.05; ** *p* < 0.01. Level of significance in univariate ANOVA (two-way analysis of variance) is denoted by ns, no significance; ** *p* < 0.01.

Release Year	N Treatments	Plant Height (cm)	No. of Panicles (×10^4^ m^−2^)	Grains Per Panicle	Grain Weight (mg)
2019–2020					
1950s	−N	107.5	377.7	32.7	35.1
	+N	116.7 **	527.0 **	30.6 ^ns^	38.3 **
1960s	−N	115.9	343.7	36.1	31.7
	+N	129.0 **	491.7 **	42.4 *	33.2 *
1970s	−N	86.1	334.7	38.9	42.1
	+N	95.9 **	530.3 **	36.6 *	43.1 ^ns^
1980s	−N	99.9	364.0	39.8	42.8
	+N	109.0 **	424.7 **	41.4 ^ns^	46.6 **
1990s	−N	86.4	369.7	42.5	42.1
	+N	88.3 ^ns^	486.7 **	43.9 ^ns^	42.2 ^ns^
2000s	−N	77.4	339.3	40.8	45.4
	+N	85.0 *	440.0 **	45.0 **	45.7 ^ns^
2010s	−N	75.4	273.3	43.7	46.9
	+N	80.8 *	394.0 **	47.5 **	47.7 ^ns^
Cultivar (C)		**	**	**	**
Nitrogen (N)		**	**	**	**
C × N		ns	**	**	ns
2020–2021					
1950s	−N	110.7	401.0	32.3	36.2
	+N	115.3 *	558.3 **	30.7 ^ns^	39.8 *
1960s	−N	118.5	381.0	33.7	29.0
	+N	131.5 **	536.7 **	35.8 *	34.2 **
1970s	−N	84.0	320.3	38.0	44.0
	+N	95.2 **	544.0 **	34.9 *	43.8 ^ns^
1980s	−N	101.9	358.7	40.0	43.1
	+N	108.6 **	421.3 **	40.5 ^ns^	46.3 *
1990s	−N	82.6	379.3	40.9	42.3
	+N	85.3 *	507.0 **	44.2 **	41.4 ^ns^
2000s	−N	79.2	326.0	41.8	44.2
	+N	81.7 ^ns^	441.0 **	46.3 **	45.6 ^ns^
2010s	−N	74.0	254.7	44.7	46.6
	+N	78.2 *	376.0 **	51.4 **	48.1 ^ns^
Cultivar (C)		**	**	**	**
Nitrogen (N)		**	**	**	**
C × N		ns	**	**	ns

**Table 3 metabolites-13-00107-t003:** Statistics of differential metabolites between cultivar and N treatment groups.

HMDB Superclass	Group				
	Cultivar		N Treatment (+N vs. −N)		
	1980s vs. 1950s	2010s vs. 1980s	1950s	1980s	2010s
Lipids and lipid-like molecules	68	22	4	19	22
Phenylpropanoids and polyketides	33	20	6	4	10
Organoheterocyclic compounds	15	6	5	7	2
Organic oxygen compounds	29	14	1	20	7
Organic acids and derivatives	8	6	2	9	5
Nucleosides, nucleotides, and analogues	1	1	1	0	1
Mixed metal/non-metal compounds	1	1	0	0	0
Lignans, neolignans and related compounds	1	0	1	0	1
Benzenoids	4	3	0	0	1
Others	11	2	3	9	5
Total	171	75	23	68	54

## Data Availability

The data presented in this study are available within the manuscripts and Appendix A.

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
