# Peer review of "Metabolite Profiling of Wheat Response to Cultivar Improvement and Nitrogen Fertilizer"

_metabolites, 2023, doi:10.3390/metabo13010107_

Round 1

Reviewer 1 Report

General comments

1. The reasons for metabolomic differences under genetic improvement and N management should be described in the Introduction or Discussion section.

2. Suggest to conduct data analysis using two-way ANOVA to analyze the interactions effect of cultivar and N application, and also should discuss in the Discussion section.

3. The font is too small to read in figures except Figure 1.

4. English writing needs to improve by native speaker.

Specific comments

Line 15, release ear or release year?

Line 23-24, was from -1.9% to 96.7%? from 19.1% to 81.6%, also for other places in the manuscript.

Line 55-56, needs reference.

Line 56-57, needs reference.

Table 1, Year of release or release year, should be consistent in the maintext.

Author Response

  1. The reasons for metabolomic differences under genetic improvement and N management should be described in the Introduction or Discussion section.

R: Yes, you are right, this should be supplemented, and we added them in our revision.

  1. Suggest to conduct data analysis using two-way ANOVA to analyze the interactions effect of cultivar and N application, and also should discuss in the Discussion section.

R: Yes, we added the two-way ANOVA to analyze the interactions effect of cultivar and N as your suggestion, thank you!

  1. The font is too small to read in figures except Figure 1.

R: Thank you for your kind reminder, the font of figures had been adjusted. However, the metabolites in Fig.7 and Fig.8 have very long chemical name, increasing the size of the font is difficult.

  1. English writing needs to improve by native speaker.

R: This English writing of our manuscript has been revised by a language edit company.

  1. Line 15, release ear or release year?

R: Ok, we have replaced ‘release ear’ to ‘release year’.

  1. Line 23-24, was from -1.9% to 96.7%? from 19.1% to 81.6%, also for other places in the manuscript.

R: All ‘-’ was changed as ‘to’, thank you.

  1. Line 55-56, needs reference.

R: Ok, we have supplemented the reference.

  1. Line 56-57, needs reference.

R: Ok, we have supplemented the reference.

  1. Table 1, Year of release or release year, should be consistent in the main-text.

R: ‘Release year’ is better, all has been consistent in this paper, thank you.

Reviewer 2 Report

After reading the manuscript entitled: Metabolomic analysis reveals the physiological mechanisms behind cultivar improvement and the application of nitrogen fertilizer on wheat yields and quality, by Fulin Zhao, Yifan Wang, Jiayu Hu, Shaolei Shi, Hongyan Zhang, Yang Wang, Youliang Ye. I make some observations.

The title could be shorter and more direct and the english writing should be reviewed;

The abstract is too long, the authors should focus on the main results and the importance of this study for society;

Some of the keywords are already in the title, the authors should try to avoid repetition;

Introduction - Describes through updated references the state of the art and important concepts for the understanding of the work;

Line 100 - 70 years from 1950 to 2010, wouldn't that be 60 years?

Results (The results could be more direct and have a little less images, it is important to choose what are the most important results;

Line 194 - (Fig.1A, B)  The authors could change the appearance order of figure 1 and table 2, once the figure 1 is cited first;

Line 205 -Table 2 - The authors should explain what * means, it is not clear neither what it is nor in what it differs; probability?

Line 210 - Fig.1C-H -Why are you explaining half of a figure in a paragraph and them introducing new information before finish to completely explain this figure?

Figure 1. Yields and quality of wheat cultivar from different release years under the -N and +N     treatments from 2019-2020 (A, C, E, G) and from 2020-2021 (B, D, F, H). This explanation is confused, where are the letters in the image? Also, the -N or +N information should be in all the figures and it could be a little smaller;

Line 246 - (Fig. 6A) Why does figure 6 appears first than figure 5?

Discussion - It feels like the discussion is a little too long because in several parts, the authors mix it with parts that are more properly to be in the introduction and results. I think that the discussion could be more direct and focused on really discussion the results. Some examples where pointed in the text;

Line 332 - 337 -The evaluation of historical sets of various cereal crops has been extensively used in earlier studies to improve the data regarding genetic advance in grain yields and associated physiological traits [21]. Two major breakthroughs in breeding have contributed to the dramatic increase in average wheat yields over the last half century. First was the Green Revolution, which saw the rise of dwarf plant breeding, while the second was distant hybridization between wheat and its wild relatives [22, 23]. - This part looks more like an introduction;

Line 338- 391  -Aside from genetic factors, crop yields can also be improved through the application of fertilizers such as N [9]. Higher wheat yields were premised by the accumulation of greater aboveground dry matter, while adequate N absorption was deemed to be critical for its accumulation [33,34]. This part looks more like an introduction;

Line 365 - .....The enrichment analysis of the KEGG pathway ...A new pargraph could start from here, this one is too long;

Line 401-426 - Cut this paragraph into two;

Conclusion -The conclusion is direct, clear, and highlights very well the importance of this study;

References - The references are up to date and relevant, however, the authors need to check the formatting.

Author Response

  1. After reading the manuscript entitled: Metabolomic analysis reveals the physiological mechanisms behind cultivar improvement and the application of nitrogen fertilizer on wheat yields and quality, by Fulin Zhao, Yifan Wang, Jiayu Hu, Shaolei Shi, Hongyan Zhang, Yang Wang, Youliang Ye. I make some observations. The title could be shorter and more direct and the English writing should be reviewed.

R: Thank you for taking the valuable time to review our manuscript. We shorted this title as ‘Metabolite profiling of the response to cultivar improvement and nitrogen fertilizer of wheat‘.

  1. The abstract is too long, the authors should focus on the main results and the importance of this study for society.

R: We have shorted our abstract, and deleted some unimportant contents.

  1. Some of the keywords are already in the title, the authors should try to avoid repetition.

R: Yes, you are right. The keywords changed as ‘yield; grain quality; metabolome; wheat release year; nitrogen nutrient’.

  1. Introduction - Describes through updated references the state of the art and important concepts for the understanding of the work.

R: Done as your suggestion, we have added some updated references.

  1. Line 100 - 70 years from 1950 to 2010, wouldn't that be 60 years?

R: Eh…yes, I am wrong, you are right.

  1. Results (The results could be more direct and have a little less images, it is important to choose what are the most important results.

R: Yes, too images are in our manuscript. After discussion, we decided to delete the volcano map Fig. 4 and Fig. 5, which seemed to be redundant.

  1. Line 194 - (Fig.1A, B) The authors could change the appearance order of figure 1 and table 2, once the figure 1 is cited first.

R: Ok, the order has been adjusted as your suggestion.

  1. Line 205 -Table 2 - The authors should explain what * means, it is not clear neither what it is nor in what it differs; probability?

R: Thank you for your kind reminder, we left out the description for ‘*, ** and ns’, and added them now. Significant differences between two N treatments for the same cultivar are indicated by ns (no significance), * (P < 0.05), ** (P < 0.01).

  1. Line 210 - Fig.1C-H -Why are you explaining half of a figure in a paragraph and them introducing new information before finish to completely explain this figure?

R: The Table 2 was used to explain the trend of yield change (Fig. 1A-B), while the Fig. 1C-H described the grain quality. After your reminder, I reorganized this paragraph to ensure that Fig. 1A-H was described first.

10.Figure 1. Yields and quality of wheat cultivar from different release years under the - and +N treatments from 2019-2020 (A, C, E, G) and from 2020-2021 (B, D, F, H). This explanation is confused, where are the letters in the image? Also, the -N or +N information should be in all the figures and it could be a little smaller.

R: Yes, we have revised these issues in revision paper. Added the letter A-H in Fig.1, and the ‘-N or +N’ in all figures get smaller.

  1. Line 246 - (Fig. 6A) Why does figure 6 appears first than figure 5?

R: The Fig. 5 has been deleted, thank you.

  1. Discussion - It feels like the discussion is a little too long because in several parts, the authors mix it with parts that are more properly to be in the introduction and results. I think that the discussion could be more direct and focused on really discussion the results. Some examples where pointed in the text.

R: Ok, your comments is to the point, we revised the Discussion in a direct and concise way.

  1. Line 332 - 337 -The evaluation of historical sets of various cereal crops has been extensively used in earlier studies to improve the data regarding genetic advance in grain yields and associated physiological traits [21]. Two major breakthroughs in breeding have contributed to the dramatic increase in average wheat yields over the last half century. First was the Green Revolution, which saw the rise of dwarf plant breeding, while the second was distant hybridization between wheat and its wild relatives [22, 23]. - This part looks more like an introduction.

R: Yeh, your comments are right, we transfer them to Introduction now. And, we revised the section of Discussion.

  1. Line 338- 391. Aside from genetic factors, crop yields can also be improved through the application of fertilizers such as N [9]. Higher wheat yields were premised by the accumulation of greater aboveground dry matter, while adequate N absorption was deemed to be critical for its accumulation [33,34]. This part looks more like an introduction.

R: Transferred them to Introduction now.

Round 2

Reviewer 1 Report

No comment.

Reviewer 2 Report

The authors reviewed the comments.